# The Performance Cost of Representational Misalignment

**Sudhanshu Srivastava**
Department of Cognitive Science
UC San Diego
San Diego, CA, USA
sus021@ucsd.edu

**Ansh Soni**
Department of Psychology
University of Pennsylvania
Philadelphia, PA, USA
asoni@sas.upenn.edu

**Meenakshi Khosla**
Department of Cognitive Science
UC San Diego
San Diego, CA, USA
mkhosla@ucsd.edu

## Abstract

Several recent results suggest that brain-like computations emerge in Deep Neural Networks (DNNs) trained on naturalistic stimuli, leading to the hypothesis that shared computations between DNNs and brains arise because these representations are necessary for optimal performance. However, existing studies primarily demonstrate correlations between alignment and performance rather than establishing causality. We address this gap by proposing a representational perturbation framework that actively promotes or suppresses representational alignment during training with reference representations while maintaining task optimization. This allows us to test whether representational alignment is necessary for optimal performance or merely coincidental. We train over 60 large-scale vision models under varying alignment constraints, constructing Pareto-optimal curves that quantify the trade-off between representational alignment and task performance. Our results consistently show that models trained to minimize alignment with oracle theoretical models, pretrained networks, or brain responses achieve worse task performance than those trained to maximize alignment, providing the first causal evidence that representational alignment is functionally important rather than epiphenomenal.

## 1 Introduction

A striking phenomenon in deep learning, particularly in vision, is the emergence of representational universality: independently trained neural networks consistently converge to similar internal representations when solving the same task. This universality extends beyond artificial systems, with deep neural networks trained on visual tasks developing representations that align remarkably well with neuronal responses across the visual hierarchy, from early cortical areas to high-level regions. Similar convergence patterns emerge when comparing networks with different architectures, initialization schemes, and training procedures[4, 2, 9, 6, 8, 11], suggesting that certain representational solutions may be fundamental to optimal visual processing.

This universality raises a fundamental question about the nature of optimal neural computation. Two competing hypotheses can explain why different systems converge to similar representations. The universality hypothesis posits that similar representations emerge because they reflect fundamental computational principles necessary for solving visual tasks effectively. Under this view, representa-

tional alignment indicates shared constraints imposed by the task structure and optimization landscape. Alternatively, the coincidence hypothesis suggests that while multiple high-performing representational solutions exist, training dynamics and architectural biases systematically favor certain solutions over others, leading to apparent universality without functional necessity.

Previous work has primarily relied on correlational evidence to support the universality hypothesis. Studies demonstrate strong alignment between ImageNet-trained CNNs and neuronal responses in visual cortex [14, 7], between different network architectures trained on the same tasks [6], and between networks and oracle theoretical models from computational neuroscience. However, correlation alone cannot establish whether representational alignment is functionally necessary or merely coincidental. The critical test requires determining whether actively disrupting alignment impairs task performance.

We address this gap with a representational perturbation framework that incorporates differentiable alignment objectives into the training loss. This allows us to actively promote or suppress alignment with reference representations—oracle models, other networks, or brain responses—while optimizing task performance. By training more than 60 networks under these constraints, we map Pareto-optimal trade-offs between performance and alignment, providing a causal test of whether alignment is functionally necessary. Our findings reveal a consistent and gradual degradation of task performance as alignment decreases. This monotonic relationship between alignment and performance establishes that representational convergence reflects functional necessity rather than mere coincidence.

## 2 Methods

Our core methodology involves training networks with a composite loss function that balances task performance with alignment objectives: $\mathcal{L}_{\text{total}} = \alpha \mathcal{L}_{\text{alignment}} + \beta \mathcal{L}_{\text{task}}$, where $\alpha$ controls the strength and direction of the alignment objective. When $\alpha > 0$, the network is encouraged to align with the reference representation (maximize alignment condition). When $\alpha < 0$, the network is trained to avoid alignment with the reference representation (minimize alignment condition). When $\alpha = 0$, standard task-only training is performed (no alignment condition). This parametric control enables systematic construction of Pareto curves mapping the trade-off between representational alignment and task performance.

We control alignment with three classes of reference representations across different experimental conditions: oracle models, pretrained network representations, and brain representations. Oracle models consist of theoretical computational models that capture known principles of brain processing. For early visual processing, we use banks of Gabor filters with systematically varied orientation and spatial frequency parameters[5]. Pretrained network representations are derived from the internal activations of fixed, previously trained networks on the same visual tasks. This condition tests whether representational convergence between independently trained networks reflects functional necessity or merely coincidental similarity in training dynamics. Brain representations consist of fMRI responses recorded from human visual cortex, specifically early visual area V1 and high-level ventral visual stream regions, using the Natural Scenes Dataset [1]. This enables direct testing of whether alignment with biological neural responses is functionally necessary for optimal visual processing performance.

The alignment metric used throughout our main results is linear alignment, computed as the negative $R^2$ for a ridge regression predicting the reference representation responses from the model layer responses. We symmetrized the metric by averaging the score from the reference-to-model and model-to-reference regressions. The validation linear alignment is calculated via 5-fold cross-validation.

Training protocols differed based on the reference representation type. For oracle model and pretrained network experiments, we performed single-label object classification on ImageNet using cross-entropy loss as the task objective. For brain data experiments using the Natural Scenes Dataset, we performed multi-label object classification on the subset of MS-COCO images used in the NSD protocol. The task objective employed sigmoid cross-entropy loss applied independently to each of 80 object classes, reflecting the multi-object nature of natural scene images. The Adam optimizer was used for each network, with a learning rate of 1e-4, with batches of size 1000.

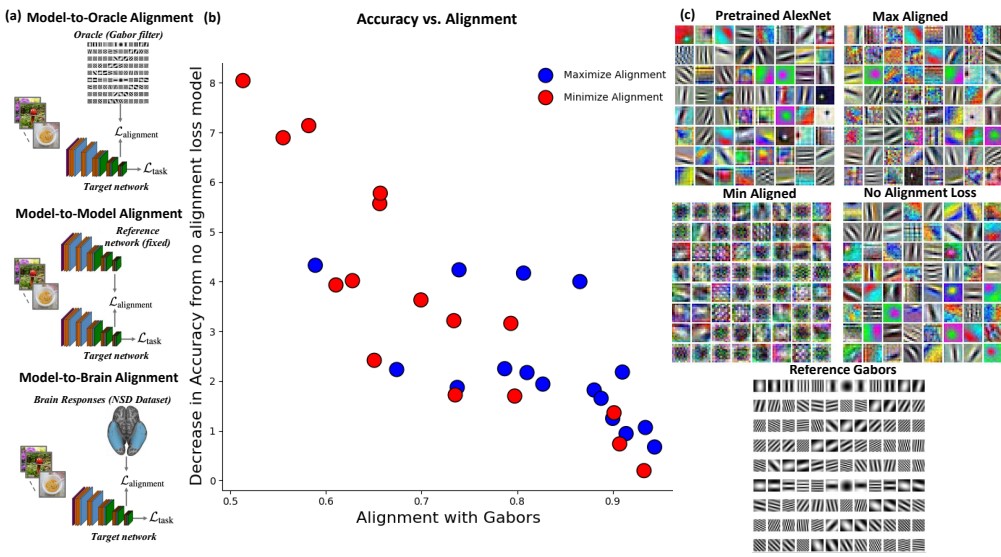

Figure 1: (a) Schematic illustration of the three alignment conditions tested: alignment with oracle models, alignment with pretrained networks, and alignment with brain representations, (b) Performance degradation relative to the no-alignment baseline as a function of oracle alignment strength. Each dot depicts the results for a different $\{\alpha, \beta\}$ pair, which controls the strength of alignment penalty. (c) Learned convolutional filters from conv1 across conditions:pretrained AlexNet, maximally aligned model, minimally aligned model, no-alignment model, and the reference Gabor filter bank used for alignment control.

## 3 Results

### 3.1 Alignment with oracle Gabor filters

We trained Alexnets [10] to align the first convolutional layer with 108 Gabor filters modeling early visual cortex selectivity. The oracle set contained 12 orientation values (15° spacing) and 9 spatial frequencies (0.05-2.0 cycles/pixel) (Figure 1a). Networks were trained for 50 epochs on ImageNet classification while simultaneously maximizing or minimizing alignment with this oracle representation. Separate networks were trained for each combination of $\alpha$ and $\beta$ in $\{1, 2, 4, 8\}$.

We observed a monotonic relationship between oracle alignment and task performance. As alignment with the Gabor filter bank decreased, ImageNet accuracy declined gradually and consistently, with the most misaligned networks showing performance drops of up to $8\%$ compared to maximally aligned models (Figure 1b). A model trained without any alignment loss performs slightly higher than models with maximized alignment.

Filter visualization revealed that aligned networks developed structured, low-to-medium frequency selective and oriented filters resembling those of standard trained networks, while progressively misaligned networks produced increasingly disorganized filter patterns (Figure 1c). This supports the hypothesis that oracle-aligned representations reflecting known cortical processing principles (e.g. orientation selectivity) are functionally necessary for optimal visual recognition.

### 3.2 Alignment with pretrained CNNs

We trained AlexNets to *promote* or *suppress* alignment with a fixed pretrained AlexNet, targeting either Conv1→Conv1 or Conv5→Conv5, for 50 epochs. Across both layers, promoting alignment yielded higher task accuracy than suppressing alignment. Sweeping the regularization strength $\alpha \in \{1, 2, 4, 8\}$ (with $\beta = 1$) revealed a graded trade-off: accuracy systematically tracked the achieved alignment with the pretrained model (Fig. 2a–b).

The performance cost of misalignment was larger when perturbing Conv1 than Conv5, indicating greater sensitivity to representational disruption in early features. This pattern is consistent with prior

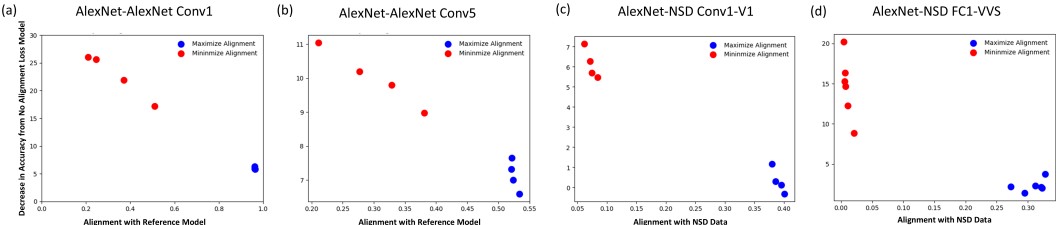

Figure 2: (a-b) Performance degradation relative to no-alignment baseline as a function of alignment strength for pretrained network experiments: Conv1-Conv1 alignment (a) and Conv5-Conv5 alignment (b). (c-d) Performance degradation relative to baseline as a function of alignment strength for brain data experiments: Conv1-V1 alignment (c) and FC1-VVS alignment (d). Each dot corresponds to a different value of $\alpha$.

observations that deeper layers can differ more across high-performing models [6], implying that some degree of cross-model misalignment at later stages coexists with strong performance.

### 3.3 Alignment with NSD responses

Our third set of experiments aligned AlexNet representations with human brain responses from the Natural Scenes Dataset (NSD; [1]), which contains fMRI recordings from eight participants viewing images from MSCOCO [13]. We focused on two regions: primary visual cortex (V1) and higher ventral visual stream (VVS) areas. Models were trained on 61,780 images with an 80/20 train–test split. Performance was measured as the mean average precision across 80 categories. All models were trained for 100 epochs.

We trained separate networks to either promote or suppress alignment between Conv1 and V1 responses ($\alpha \in \{0.1, 0.2, 0.4, 0.8\}, \beta = 1$) and between FC1 and VVS responses ($\alpha \in \{0.01, 0.02, 0.04, 0.08, 0.1, 0.2\}, \beta = 1$).

Consistent with the results for the oracle and pretrained models, we find that aligned networks perform better than misaligned networks for both Conv1-V1 (Figure 2c) and FC1-VVS (Figure 2d). For Conv1-V1 alignment, systematic reduction in alignment strength produced graded performance decreases, with complete misalignment resulting in a 7-point drop in mean average precision compared to baseline (Figure 2c). The FC1-VVS alignment condition showed an even more pronounced effect, with minimally aligned networks suffering performance costs of nearly 20 mAP points relative to the no-alignment baseline (Figure 2d).

**Results for a larger network.** Besides our results for AlexNet, we also reproduce our results for ResNet-18 for aligning with the NSD dataset for Block1 and V1, and for Block4 and VVS. We used $\alpha = 0.05$ and $\beta = 1$. We again find that misaligned networks perform worse for both Block1-V1 and Block4-VVS, and the effect is larger for Block4-VVS (Figure A.1).

## 4 Discussion

Our results provide the first causal evidence that representational alignment across different computational systems is functionally important rather than coincidental. While previous studies have looked at the effect of maximizing alignment [12, 3], the effect of minimizing alignment has not been studied before. The consistency of effects across theoretical models, pretrained networks, and biological neural responses demonstrates that representational universality reflects fundamental computational principles necessary for optimal visual processing.

The representational perturbation framework offers several methodological advantages for future research. It enables causal inference by moving beyond correlational studies to test functional necessity. The framework provides systematic evaluation through parametric study of alignment-performance trade-offs via Pareto curve construction. It demonstrates general applicability by working with any differentiable alignment metric and proves scalable to large-scale training scenarios. This

approach opens new directions for understanding computational principles underlying intelligent processing across artificial and biological systems.

**Limitations.** Our findings may show architecture specificity, as effects could vary for fundamentally different network designs beyond the CNN architectures tested. Task dependence represents another consideration, since alignment necessity may vary across different visual tasks or domains. Additionally, our framework may interact with optimization dynamics in complex ways that require further investigation.

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

# A    Technical Appendices and Supplementary Material

(a)         ResNet-NSD Block1-V1                    (b)         ResNet-NSD Block4-VVS

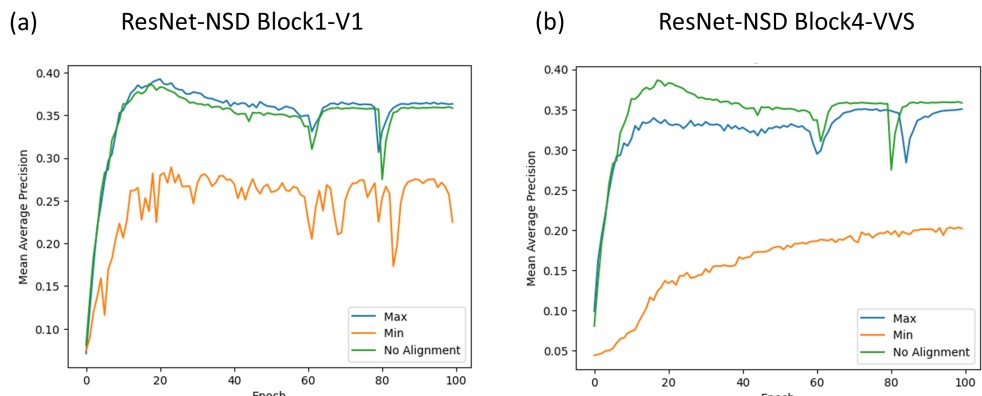

Figure A.1:  Epoch-wise mean average precision for ResNet-18 trained with maximizing, minimizing, and no alignment loss for linear alignment for Block1-V1 (a) and Block4-VVS (b).

