# OpenReview forum: "The Performance Cost of Representational Misalignment"
_NeurIPS.cc/2025/Workshop/UniReps — UniReps2025_

### Official Review · Reviewer_Xb99 · 2025-09-11
**Testing the connection of representational alignment and network functions**

**Confidence:** 4

**Review:**

**Summary:** This paper tests if representational alignment between different artificial and biological neural networks is important for network function or if alignment is just epiphenomenal. The authors train networks to minimise a composite loss that consists of a term that either maximises or minimises alignment and a term for normal task loss. The authors show correlations between model alignment and loss in task performance.

**Strengths:**
- The paper tackles an important problem. Linking representational universality that has emerged across artificial and biological neural networks to the functional role that these representations play in network behaviour. I think the idea to causally manipulate alignment is the right way to dissect if alignment is epiphenomenal or indeed functionally important.
- The approach to have a composite loss is a valid approach to this problem and is a technically sound way to test this.
- I think in general the results support the authors claims that lower representational alignment is linked to decreases in model accuracy.
- I think the theme of the paper is a perfect fit for the theme of the workshop.

**Weaknesses:**
- For figure 1. It is a bit confusing that there is so much variation for models that are trained to maximise and minimise alignment. I would have liked to see an analysis that explains why some models that are literally trained to minimise alignment are still highly aligned. I assume it relates to the ratio of  \alpha and \beta.
- The figures could be improved. In Fig 1, b) the specific values of \alpha and \beta are not highlighted. This makes it difficult to discern how strongly alignment and accuracy interact with the various settings of the optimisation objective. Also what setting of \alpha and \beta were used in Fig 1, c)?
- Relatedly, would it not make more sense in general to represent the different settings of \alpha and \beta as a ratio. This value would represent how strongly the two terms in the loss function are in competition.
- Similarly, as a control I would have liked to see what is the average alignment with Gabor filters that can be expected when training a model without any alignment loss.
- I also am concerned about the fixed length of training (50 epochs). How was this chosen? Did all models converge at this point? Is it possible that models trained to maximise misalignment will still achieve high accuracy later in training. I think that part of the decrease in performance could be ascribed to changes to the loss landscape. Maybe good minima are harder (but not impossible) to find in these cases?
- Finally, I think the paper could also be enhanced by an alternative experimental setting which would corroborate the author's claims. Start from a pre-trained model and try to train the model on a composite loss that minimises alignment to a pre-trained reference model while attempting to keep behaviour equal. If this is very hard or impossible this would also strengthen the author's claims.

**Score:**

3

**Topic Fit:**

3

---

### Official Review · Reviewer_qPdJ · 2025-09-12
**Well-executed alignment study with major analytical issues.**

**Confidence:** 4

**Review:**

This work is line with the the interests of the workshop. The authors explore the effects of alignment and misalignment with three classes of relevant reference representations and report mostly clear results. However, there are significant concerns that should be addressed in order for the paper to warrant acceptance.

1) It is unclear how the results support the conclusion. If it is possible that the previously observed correlation between alignment and performance is due to "shared training dynamics and architectural biases", how is the coincidence hypothesis disproved by experiments that don't alter the fundamental training dynamics or architectural biases of the networks being trained? It seems entirely possible that the misalignment objective discourages networks from achieving the representations that their architectures and training dynamics require to achieve high task performance, which would not disqualify the coincidence hypothesis.

2) For the most part, both alignment and misalignment degrade task performance relative to no alignment. However, previous works have demonstrated that including alignment objectives can actively increase task performance, such as "Improved object recognition using neural networks trained to mimic the brain’s statistical properties". It is difficult to make sense of the relative impact of alignment and misalignment when the alignment itself does not seem to be working to the extent the literature suggests that it should. This also calls the framing of the paper (that alignment is positively correlated with task performance) into question. Additionally, the use of Pareto-optimal curves to analyze the experiments demonstrates that the authors assumed that the two objectives would be competing, which, again, should not necessarily be the case.

3) Prior works have already shown a causal link between alignment and performance in computer vision by offering explanatory frameworks for *how* alignment leads to better performance. Namely, "Robust deep learning object recognition models rely on low frequency information in natural images", shows that aligned networks focus on coarse visual information which improves robustness to image corruptions and over dependence on small scale information that may correspond to multiple classes of objects. At this point in time, any work that claims to establish a causal link between alignment and performance without offering novel insights into the causal mechanism of this relationship lacks the necessary novelty required to be published in the neural alignment literature.

As a minor critique, only experiments with Beta = 1 are reported in the paper. For clarity, it would be best to remove all mention of this parameter from the paper and simply present the loss as L_task + alpha * L_alignment.

While the critiques listed above are significant, I am open to changing my score through discussion :)

**Score:**

1

**Topic Fit:**

3

---

### Official Review · Reviewer_9eHf · 2025-09-15
**Causal Evidence for the Functional Necessity of Representational Alignment in Neural Networks**

**Confidence:** 4

**Review:**

Quality:
- Very sound methodology: trained 60 networks under controlled restraints, actively manipulating representational alignment to test whether alignment is functionally necessary for task performance
- Experiments effectively isolate alignment effects from task optimization and include various reference types (oracle theoretical models, pretrained networks, and human brain responses), allowing causal conclusions about alignment
- Results are visualized and presented clearly with filter analyses and graded performance curves, showing that changes in alignment lead to predictable changes in performance
- Limitations such as task dependence and architectural specificity to the CNN architectures tested are acknowledged
- Only linear alignment metrics were used; other measures such as CKA may yield alternative insights
- Interactions between alignment constraints and training dynamics could have been explored further

Clarity:
- Could benefit from more formal definitions of alignment metrics (to maximize technical precision)
- Academic writing is clearly structured and progresses logically
- Figures visualize and illustrate alignment-performance relationship, making causal effects observable

Originality and Significance:
- Prior work relied on correlation between network representation and brain data, whereas this paper manipulates alignment directly to provide causal evidence
- Provides effective evidence that representational alignment is functionally necessary, not merely correlated with task performance
- Insights and findings can be further applied to deep learning, though this was not explored in the paper

Overall Pros:
- Provides causal evidence of the functional necessity of representational alignment through direct manipulation experiments
- Sound methodology with relatively large experimental design supporting causal inference
- Tests alignment across oracle models, pretrained networks, and human brain responses
- Strong relevance to workshop themes on representational universality

Cons:
- Limited to CNN architectures (AlexNet, ResNet-18), meaning generalization to other architectures was left untested
- Only linear alignment metrics were used; other measures such as CKA may yield alternative insights
- Potential applications to deep learning are suggested but not fully explored
- Significance in deep learning is mentioned but not demonstrated

**Score:**

4

**Topic Fit:**

3

---

### Official Review · Reviewer_gv8r · 2025-09-16
**The paper contributes a novel causal framework showing that representational alignment is functionally necessary, despite overlaps with prior work and limited experimental scope.**

**Confidence:** 3

**Review:**

## Strength
**S1. Strong clarity and coherence, making its contributions accessible**
- The paper is well-written and follows a logical structure, progressing smoothly from introduction through methods, results, and discussion. The use of visualizations such as filter comparisons and Pareto curves reinforces the main arguments and makes the findings more interpretable.
- The proposed method is well-founded, employing controlled training losses with systematic variation of alignment coefficients. By incorporating oracle models, pretrained networks, and brain data, the study achieves broad coverage that increases the robustness of its conclusions.

**S2. Offering meaningful insights into the causal role of representational alignment**
- The proposed framework can introduce causal perturbation of alignment, examining both positive and negative alignment objectives. This goes beyond prior correlational work, enabling stronger claims that alignment is not merely correlated with performance but functionally necessary.
- The construction of Pareto frontiers provides a systematic mapping of alignment strength against task performance. Unlike prior studies that typically report single regularization outcomes, this approach reveals the full spectrum of trade-offs. Furthermore, the findings are consistent across diverse reference targets, including Gabor filters, pretrained CNNs, and human brain responses.

## Weakness

**W1. Overlap with prior work**
- The concept of regularizing models toward biological neural representations has already been introduced in prior research [1]. In the work, a specialized regularization term penalizes dissimilarity between model features and neural responses, ensuring that image representations approximate biological activity during training.

**W2. Limited Experimental Setup and Analysis**
- The study primarily evaluates AlexNet, which restricts the breadth of conclusions. In [2], prior comparisons between various CNNs have shown that, beyond convolutional biases, factors like dataset composition and training procedures substantially influence representational alignment. Given AlexNet’s shallow design and strong locality bias, it struggles to capture long-range dependencies or global shapes early, unlike modern models with skip connections, multi-scale receptive fields, and attention mechanisms. As a result, the performance degradation under misalignment observed here may not generalize to architectures with greater representational flexibility.
- Furthermore, the reported results lack statistical rigor, with no error bars or significance testing provided for some plots. This absence raises the possibility that the outcomes reflect architecture-specific biases rather than universally applicable principles.

[Reference]
- [1] Li, Zhe, et al. "Learning from brains how to regularize machines." Advances in neural information processing systems 32 (2019).
- [2] Conwell, Colin, et al. "A large-scale examination of inductive biases shaping high-level visual representation in brains and machines." Nature communications 15.1 (2024): 9383.

**Score:**

3

**Topic Fit:**

3